# Synthesis of arylamines and N-heterocycles by direct catalytic nitrogenation using N$_2$

Kai Wang [1], Zi-Hao Deng[1], Si-Jun Xie[1], Dan-Dan Zhai[1], Hua-Yi Fang [1] & Zhang-Jie Shi[1,2✉]

Ammonia and nitric acid are two key platform chemicals to introduce nitrogen atoms into organic molecules in chemical industry. Indeed, nitric acid is mostly produced through the oxidation of ammonia. The ideal nitrogenation would involve direct use of dinitrogen (N$_2$) as a N source to construct N-containing organic molecules. Herein, we report an example of direct catalytic nitrogenation to afford valuable diarylamines, triarylamines, and N-heterocycles from easily available organohalides using dinitrogen (N$_2$) as the nitrogen source in a one-pot/two-step protocol. With this method, $^{15}$N atoms are easily incorporated into organic molecules. Structurally diversified polyanilines are also generated in one pot, showing great potential for materials chemistry. In this protocol, lithium nitride, generated in situ with the use of lithium as a reductant, is confirmed as a key intermediate. This chemistry provides an alternative pathway for catalytic nitrogenation to synthesize highly valuable N-containing chemicals from dinitrogen.

[1] Department of Chemistry, Fudan University, Shanghai 200433, China. [2] State Key Laboratory of Organometallic Chemistry, Shanghai Institute of Organic Chemistry, Chinese Academy of Sciences, Shanghai 200032, China. ✉email: zjshi@fudan.edu.cn

As known, nitrogen is a key component of nucleosides and proteins, which form the basis of life, allowing genetic information to be passed through species and sustaining vital activities on earth[1]. N-containing molecules in drug design and polymeric materials greatly promote the quality of human beings' life[2,3]. Nitrogen is also an indispensable element in most highly energetic materials[4], which have been broadly used in military, aeronautical, and space technologies. Recently, various new N-containing molecules have been designed and synthesized, showing their utilities as photoelectronic materials[5]. Owing to the importance of N-containing molecules, the incorporation of nitrogen atoms into organic molecules through C–N bond formation is a major challenge in organic chemistry[6].

As a long-term fundamental challenge, the activation and direct transformation of dinitrogen ($N_2$) has received much attention since the early twentieth century[7]. The transformation of dinitrogen into ammonia through heterogeneous catalysis has a long history of application (Fig. 1a), and has been recognised as the most important reaction for promoting human's life[8]. Recent advances have demonstrated the potential of homogeneous catalysis to produce ammonia directly from dinitrogen using complex catalysts in the presence of highly valuable reductants and proton sources[9]. With great efforts by coordination chemists, various complexes with dinitrogen ligands have been prepared using almost all transition metals in different coordination modes in the last half century[10,11]. By Coordination, dinitrogen in complexes was activated and has been used to prepare different N-containing organic compounds in a stoichiometric manner (Fig. 1b)[12–22]. Despite some synthetic cycles having been reported

for the synthesis of N-containing molecules[23–26], to the best of our knowledge, catalytic transformation to carry out direct nitrogenation of organic molecules from dinitrogen ($N_2$) has not been approached up to date. Moreover, the use of active organoelectrophiles and strong reductants in synthetic cycles greatly limits their applications. On the other hand, ammonia and nitric acid, mostly produced through ammonia oxidation, are two key platform chemicals to introduce nitrogen atoms into organic molecules in chemical industry (Fig. 1a). So far, there are two major methods applied to producing N-containing organic compounds industrially: (i) Nitration using nitric acid (produced through ammonia oxidation)[27], and (ii) C–N coupling and nucleophilic substitution using ammonia and its derivatives[28]. In this study, we aimed to produce organic compounds directly from dinitrogen by avoiding traditional nitrogenation processes based on platform chemicals, ammonia, and nitric acid. After reviewing the state-of-the-art nitrogenation using dinitrogen, we conceived that alkali or alkaline-earth metals, which are relatively cheap and easily available, might be feasible reductants to reach this goal[29]. As organohalides are abundant and have been widely used in C–N bond formation[30], we attempted to apply them as general electrophiles to construct C–N bonds through transition-metal-catalyzed C–N coupling.

The use of alkali or alkaline-earth metals, such as lithium (Li) and magnesium (Mg), as reductants for our designed direct nitrogenation is possible owing to their high reactivity toward unreactive dinitrogen. For example, Mg can burn in pure dinitrogen to produce magnesium nitride under critical conditions[31]. Lithium nitride ($Li_3N$) was also prepared from lithium and dinitrogen at 400–500 °C[32,33]. Moreover, aryllithium (ArLi) was shown as an interesting starting point for direct nitrogenation with dinitrogen via Ti catalysis to produce aniline derivatives (Fig. 1c), representing the first successful catalytic transformation in amine synthesis, albeit the efficiency was not sufficient for applications[34,35]. Another beautiful example in the field of nitrogenation reported by Mori disclosed a catalytic transformation using aryl halides and dinitrogen to produce diaryl/monoarylamines (affording a mixture in most cases) through co-catalysis by Pd and Ti complexes (Fig. 1c)[36,37]. The relatively complicated conditions made the mechanism arguable, and the roles of Li and both transition-metal complexes were difficult to be determined at this stage. We envisaged that, under appropriate mild conditions, dinitrogen might be transformed into organic compounds by reacting with various electrophiles through transition-metal catalysis (Fig. 1d). The use of alkali or alkaline-earth metals to in situ reduce dinitrogen may be a good choice. This design would provide an efficient approach to N-containing molecules from dinitrogen by avoiding the tedious and energy-consuming procedures required by ammonia and nitric acid as platform chemicals.

Here, we show an example of direct catalytic nitrogenation to produce valuable nitrogen-containing organic molecules and polymers from easily available organohalides with dinitrogen ($N_2$) as nitrogen source in one-pot/two-step protocol. In our studies, biarylamines, triarylamines, and heterocycles are synthesized through in situ-generated $Li_3N$ as a key intermediate, showing its application for nitrogen-containing organic chemicals.

**a Conventional Nitrogenation**

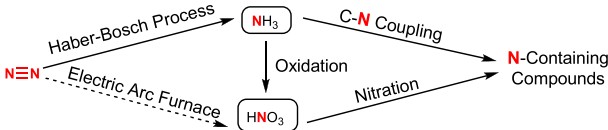

**b Nitrogenation through Transition-metal [M-N] Complexes**

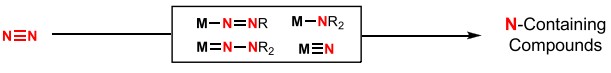

**c Direct Nitrogenation based on [Ti-N] Complexes from N₂**

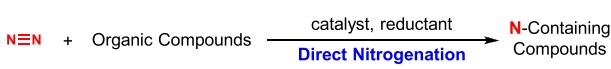

**d Direct Nitrogenation from N₂ (Our Target)**

**Fig. 1 Desired direct nitrogenation of organic compounds using dinitrogen (N₂). a** Traditional nitrogenation procedures using $NH_3$ and $HNO_3$ as platform chemicals. **b** Production of N-containing organic compounds from dinitrogen using M–N complexes in stoichiometric reactions. **c** Incorporation of $N_2$ into organic compounds with stoichiometric Ti–N complexes as intermediates. **d** Desired catalytic nitrogenation using $N_2$.

## Results

**Hypothesis**. Based on our hypothesis, we conducted the direct nitrogenation of *o*-tolyl bromide (**1a**) using dinitrogen in the presence of Li as reductant and $Pd_2(dba)_3$ as catalyst in dioxane. Various ligands were screened, and we found that RuPhos was efficient to afford the desired di(*o*-tolyl)amine (**2a**) in 17% isolated yield. It is important to note that the desired nitrogenation was achieved and a valuable diarylamine was produced from

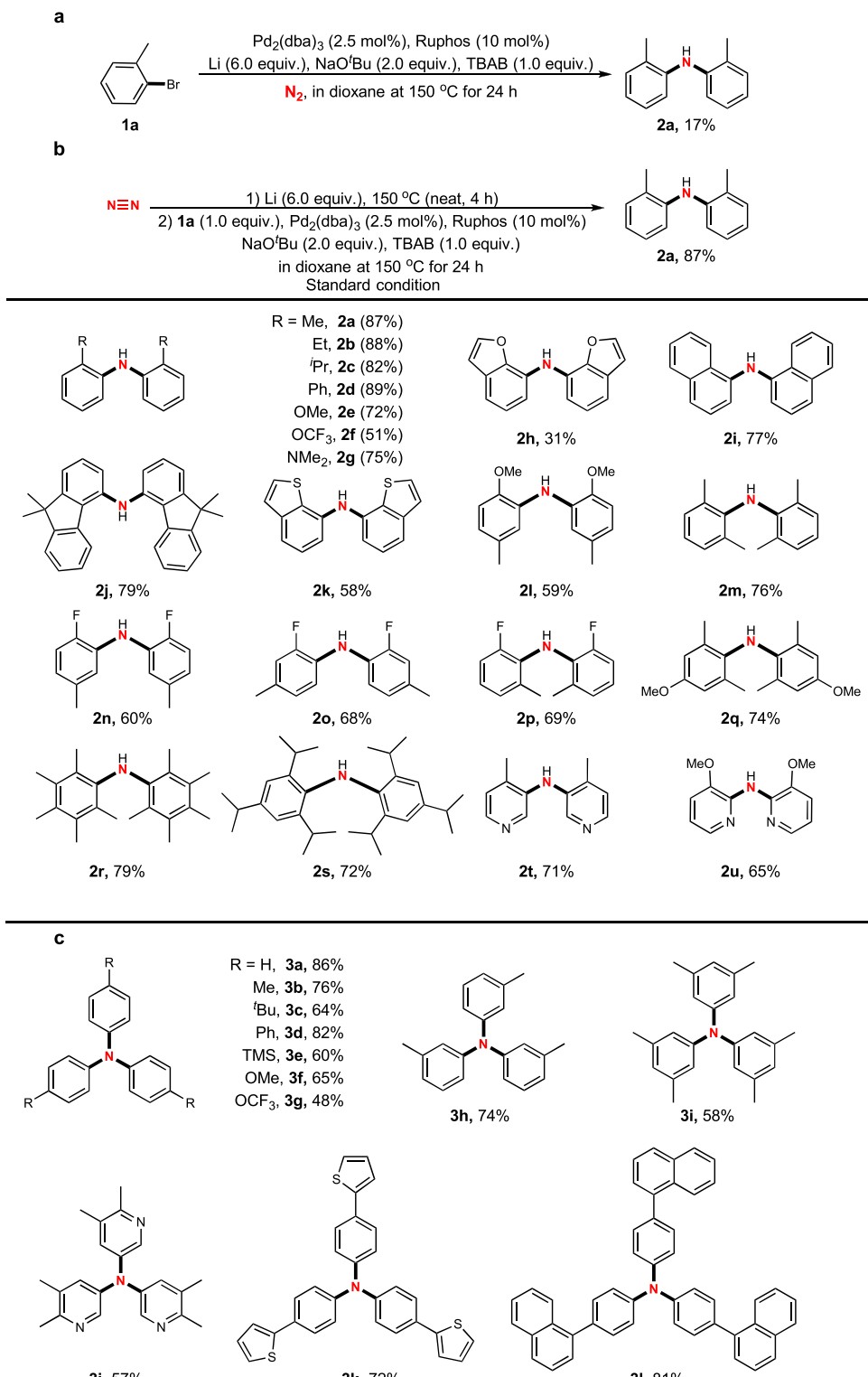

**Fig. 2 Direct nitrogenation using dinitrogen (N₂) to synthesize bi/triarylamines. a** Catalytic C–N formation using dinitrogen (N₂) to synthesize bi(o-tolyl) amine. **b** One-pot/two-step nitrogenation for the synthesis of diarylamines directly from ortho-substituted aryl bromides using dinitrogen (N₂). **c** One-pot/ two-step nitrogenation to synthesis triarylamines directly from meta/para-substituted aryl bromides using dinitrogen (N₂).

dinitrogen in a catalytic manner. Further efforts failed improving the yield, and the reduced byproduct toluene was always obtained (Fig. 2a). We proposed that the formation of the desired product proceeded through in situ-generated Li₃N as a key intermediate, followed by a Pd-catalysed C–N coupling reaction. The kinetic

reaction rate between the solid phase (Li) and gas phase (N₂) in solution was rather slow, which was not comparable to the reaction between the Li solid and the solution of ArBr through Li–halogen exchange. Therefore, toluene was obtained as a major byproduct after the protonation of aryllithium.

**Optimization of reaction conditions**. According to the rational analysis, we conceived that a one-pot/two-step protocol might solve this problem. A procedure was performed, with the solid–gas bi-phase reaction conducted in a vessel containing Li powder in $N_2$ atmosphere at 150 °C for 4 h. Subsequently, a solution of o-tolyl bromide in dioxane with $Pd_2(dba)_3$ (2.5 mol%) and RuPhos (10 mol%) as a catalyst set was injected into the vessel. In the presence of $NaO^tBu$ as a base and tetrabutylammonium bromide as a phase transfer reagent, the desired product **2a** was obtained in 87% isolated yield (Fig. 2b). This is a successful and valuable catalytic transformation for preparing N-containing organic compounds directly from dinitrogen, which exhibited a great potential for the synthesis of useful diarylamines.

**Scope of aryl bromides**. We further tested different o-substituted aryl bromides under the standard conditions (Fig. 2b). To our delight, alkyl groups other than methyl at the *ortho*-position of aryl bromides did not affect the efficiency, although the sterically hindered isopropyl group slightly decreased the yield (**2b–2c**). Phenyl substituent was tolerated (**2d**). The less-hindered OMe group also controlled the selectivity to produce **2e**. However, the electron-withdrawing $OCF_3$ group dramatically diminished the yield (**2f**). Fused-ring derivatives were also successfully introduced into this system and more complicated diarylamines were isolated in moderate to good yields, no matter whether a heteroatom was present or not in ring system (**2h–2k**). In the presence of *ortho*-substituents, additional substituents were also introduced at other positions and the corresponding diarylamines were obtained as sole products, regardless of whether di-, tri-, and even penta- substituents present (**2l–2r**). Notably, this nitrogenation was compatible with high steric hindrance. For example, 2,6-dimethyl substituted phenyl bromides were also suitable substrates and the products were isolated in high efficacy (**2m, 2q,** and **2r**). To our interest, a sterically bulky 2,4,6-diisopropyl phenyl bromide was submitted and the desired product **2s** was also afforded. Although a fluoro substituent is considered to be relatively small, it still controlled the chemo-selectivity to obtain diarylamine **2n**. Moreover, heterocyclic bromides were also compatible with this nitrogenation. Both 2-bromopyridine and 3-bromopyridine derivatives with different *ortho*-substituents were applied to this nitrogenation, and the desired products (**2t** and **2u**) were isolated in good yields.

To our interest, when substituents were absent or present at *meta*- and/or *para*-positions of aryl haides, triarylamines were selectively produced owing to the lower steric hindrance (Fig. 2c). Electron-donating groups were better tolerated than electron-withdrawing ones (**3a–3f** vs **3g**). Trimethyl silyl group was also well tolerated, providing the potential for further orthogonal functionalization of **3e**. Both *meta*- and *para*- tolyl groups gave comparable yields (**3b** and **3h**). Heterocycles, such as substituted pyridine and thiophene derivatives, showed credible reactivity to produce tertiary amines in good yields (**3j** and **3k**).

**Synthetic applications**. Starting from 2,2'-dibromobiphenyl derivatives **4**, carbazole derivatives were produced as a sole product in good to excellent yields (Fig. 3a). In comparison, ligand $Ad_2P^nBu$ was found most efficient to promote this reaction. Starting from 2,2'-dibromodiphenyl **4a**, desired product **5a** was produced in 78% isolated yield. The presence of a methyl group did not affect the efficiency. Other functionalities, such as $OCF_3$ and F, were compatible with this nitrogenation, regardless of their positions. This chemistry provided an effective alternative method for producing functionalised carbazoles using dinitrogen directly in one-pot/two-step protocol.

We further conducted a substitution reaction using the one-pot/two-step procedure, and the amine was produced in good yield as predicted. For example, when benzyl bromide **6** was used as the electrophile, tribenzylamine **7** was prepared from nitrogen in 81% isolated yield. Using this developed method, polysubstituted pyrrole **9** was also prepared directly from dinitrogen and 1,4-dibromo-1,3-butadiene **8** in good efficiency, further extending the application of this chemistry (Fig. 3b).

Since diarylamines, triarylamines, and carbazoles have been broadly applied in materials chemistry and drug discovery[38–40], we next explored the application of this nitrogenation to the synthesis of complicated structures to explore potential applications (Fig. 3c). From commercial bromide **10**, corresponding diarylamine **11** was produced in 63% isolated yield. From terphenyl bromide **12**, the corresponding triterphenylamine **13** was prepared by direct nitrogenation in 89% isolated yield. This compound showed excellent properties as a hole-transporting material[41]. Furthermore, as reported above, carbazole **5a** was prepared from 2,2'-dibromodiphenyl using this nitrogenation protocol. After nucleophilic substitution, N-(p-bromophenyl)-carbazole **14** was obtained in excellent yield[42]. Using this standard nitrogenation procedure, complex triarylamine **15**, which has also been successfully used as a hole-transporting material[43], was prepared in 77% isolated yield. In this synthesis, all four nitrogen atoms in the molecule came directly from dinitrogen, confirming the great potential applications of this chemistry in materials synthesis.

As [15]N-labelled molecules have been used to monitor bioactivity in biosystems[44], we further expanded our chemistry to incorporate [15]N atoms into privileged scaffolds in drug discovery (Fig. 3d). As this developed protocol was easy to handle, [15]N atoms were readily incorporated into the desired molecules when [15]$N_2$ was used instead of [14]$N_2$. When **4a** was used, [15]N-incorporated carbazole was prepared in 76% isolated yield. When 2,2'-dibromodiphenyl sulfide **16** was used, the desired [15]N-incorporated 10H-phenothiazine **17** was produced in 63% isolated, which is the core structure of commercial drug promethazine[45].

Polyanilines (PANs) are important materials with broad applications in electron field emission sources[46], urease sensors[47], electrode materials in transitional lithium batteries[48], and others[49]. Traditionally, PANs are readily prepared from aniline by oxidation[50]. However, this traditional method can only produce *para*-polymers in different oxidation states, including leucoemeraldine, protoemeraldine, emeraldine, nigraniline, and pernigraniline. Owing to non-requirement of redox reagents in the step of C–N formation in our system, we predicted that our method might provide an alternative pathway to prepare PANs in highly reductive states. Most importantly to note that, starting from different dihalobenzene derivatives, special site-specific PANs might be designed and synthesised using this method.

To prove this concept, we investigated the synthesis of PANs directly from $N_2$ starting from readily available dibromobenzenes as starting materials (Fig. 4). Using 1,4-dibromobenzene, 1,4-polyaniline was prepared using such a simple one-pot/two-step procedure. The molecular weight was up to 31,850 Da, as determined by gel permeation chromatography (GPC). Interestingly, when 1,3-dibromobenzene was used, the desired 1,3-polyaniline was produced with a much higher molecular weight (82,593 Da). Furthermore, when 4,4'-dibromobiphenyl was used, biphenyl-bridged polyaniline was produced with a relatively low molecular weight. It is important to note that, when less-hindered 4,4'-dibromobiphenyl was used as starting materials, triarylamine was constructed as the unit in polymer, which was detected by IR and might show unique property in material chemistry. Undoubtedly, this method proved to be most efficient for preparing bridge-diversified polyanilines, which are semiconducting materials.

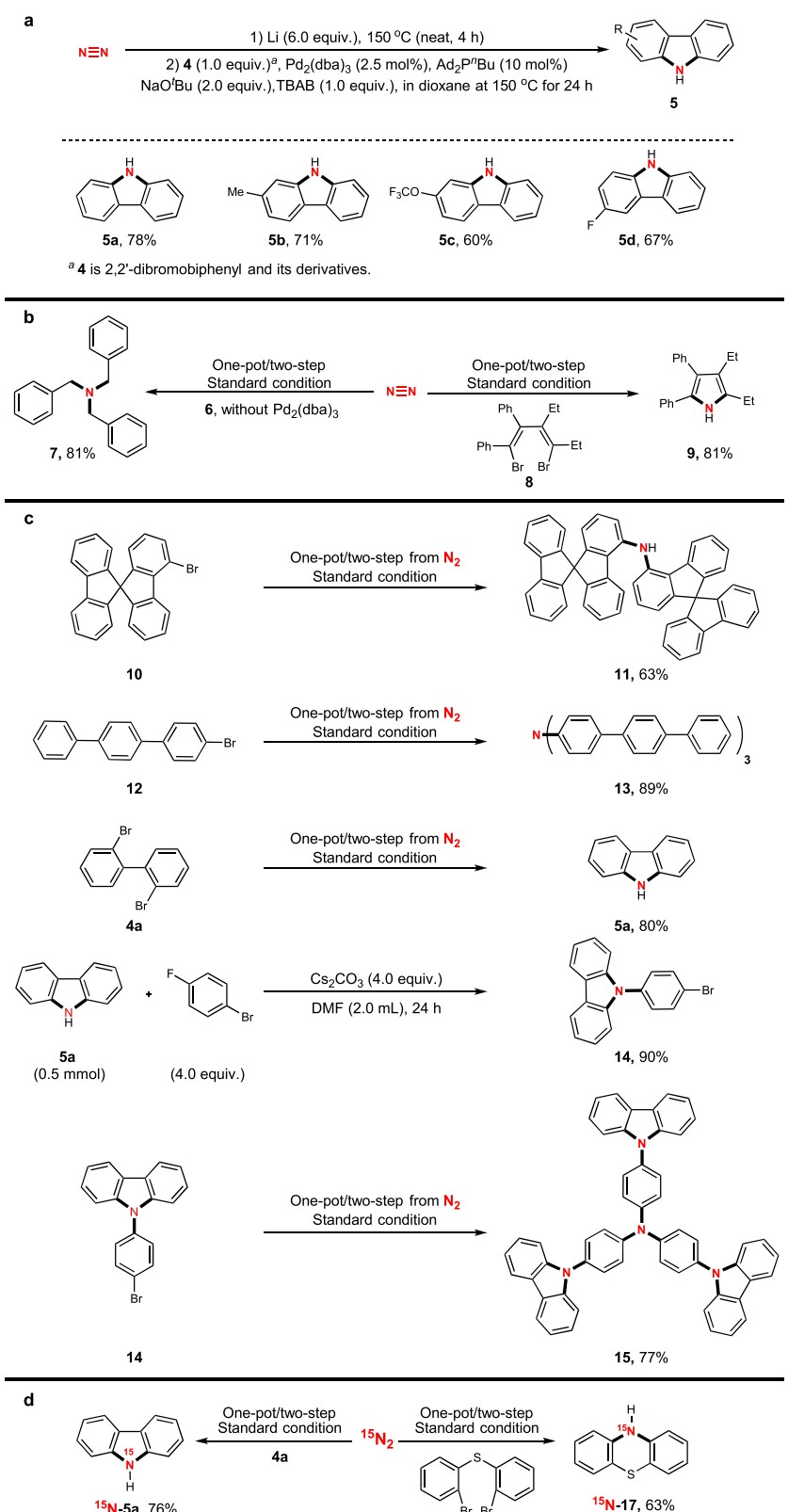

**Fig. 3 Synthetic applications. a** Carbazole synthesis through one-pot/two-step nitrogenation. **b** One-pot/two-step procedure to carry out direct nitrogenation using dinitrogen ($N_2$) to synthesize tribenzylamine and pyrroles. **c** Synthesis of complicated structures through direct nitrogenation using dinitrogen ($N_2$). **d** Synthesis of [15]N-labelling complexes through direct nitrogenation using dinitrogen ([15]$N_2$).

**Fig. 4 Synthesis of polyanilines.** Synthesis of structural diversified PANS through direct nitrogenation using dinitrogen ($N_2$).

**Mechanistic investigations**. Based on current experimental data, we proposed that this nitrogenation proceeded through lithium nitride as a platform intermediate. Usually, $Li_3N$ is prepared at 400–500 °C[32,33]. Thus, the formation of $Li_3N$ under our reaction condition at 150 °C is arguable. On the other hand, nitrides have rarely been applied to C–N bond formation[51]. To further clearly understand this nitrogenation, after the reaction of Li and dinitrogen at 150 °C for 4 h, $Li_3N$ was solidly confirmed. In fact, $Li_3N$ can been easily prepared on a large scale and commercially available in relatively low price. Starting from commercial $Li_3N$, di($o$-tolyl)amine (**2a**), triphenylamine (**3a**), and carbazole **5a** were produced from the corresponding halides and isolated in comparable yields with the one-pot/two-step process from dinitrogen directly. As the condition of the nitrogenation is relatively rough, the process might go through the heterogeneous manner. The mercury-drop experiment was conducted (for details, please refer to Supplementary Table 2). After 15 min when reaction started, excess 300 equiv. of Hg(0) (vs Pd) was added, and the reaction was then allowed to stir for the next 12 h. and we found that the yields were clearly decreased compared to the other parallel reaction without Hg(0), indicating that the process is most likely to the catalytic characteristic of palladium nanoparticles[52].

## Discussion

In summary, a one-pot/two-step protocol for direct catalytic nitrogenation by using dinitrogen as nitrogen source is developed through Pd catalysis in the presence of Li as reductant. This chemistry was proved to occur through $Li_3N$ as the key platform chemical. This is also an example of $Li_3N$ being used to construct C–N bonds in organic synthesis. Not only does the method show that $Li_3N$ is a potential platform chemical for the synthesis of N-containing molecules, but also allows direct nitrogenation to obtain N-containing molecules from dinitrogen while avoiding the traditional $N_2$-$NH_3$-$HNO_3$ procedure. However, the chemistry is still suffering from the relative harsh condition and the use of Li as reductant. Further efforts to explore other cheaper and safer reductants and expand the synthetic applications of useful chemicals are underway in our laboratory.

## Methods

**General procedure of one-pot/two-step for direct catalytic nitrogenation using $N_2$.** In an argon-filled glovebox, a 25 mL oven-dried seal-tube equipped with a magnetic stir bar was charged with lithium powder (6.0 equiv., 3.0 mmol, 0.0210 g). The tube was removed from the glovebox, degassed, and refilled with $N_2$. The tube was stirred under nitrogen atmosphere at 150 °C for 4 h. Then aryl bromide (1.0 equiv., 0.50 mmol), tris(dibenzylideneacetone)dipalladium (0.025 equiv., 0.0125 mmol, 0.0114 g), 2-dicyclohexylphosphino-2′,6′-diisopropoxybiphenyl (0.10 equiv., 0.05 mmol, 0.0233 g), sodium tert-butoxide (2.0 equiv., 1.0 mmol, 0.0961 g), tetrabutylammonium bromide (1.0 equiv., 0.50 mmol, 0.1610 g) and dioxane (2.0 mL) were added into the above tube, and the reaction mixture was stirred under nitrogen atmosphere at 150 °C for 24 h. The reaction mixture was allowed to cool to room temperature and quenched by water. A saturated solution of brine and ethyl acetate was added. The aqueous phase was extracted three times with ethyl acetate. The combined organic layer was washed with brine and filtered. The solvent was removed under reduced pressure. Finally, the residue was purified by flash chromatography on silica gel.

## Data availability

All data that support the findings of this study are available within this article and its Supplementary Information (including experimental procedures and compound characterization data). Data are also available from the corresponding author upon reasonable request.

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

## Acknowledgements
We are grateful to the National Natural Science Foundation of China (no. 21988101, 21761132027), Science and Technology Commission of Shanghai Municipality (19XD1400800, 18JC1411300), Shanghai Municipal Education Commission (2017-01-07-00-07-E00058), Key-Area Research and Development Program of Guangdong Province (2020B010188001) for support of this research. We thank Simon Partridge, PhD for editing the English text of a draft of this manuscript.

## Author contributions
S.Z. directed the research and developed the concept of the reaction with W.K., who also performed the experiments and characterized all the products. D.Z., Z.D., X.S., and F.H. gave some helpful suggestions for the reaction. S.Z. wrote the manuscript with contributions from the other authors.

## Competing interests
The authors declare no competing interests.
