## [Peer Review File · Nature Communications]

Reviewers' Comments:

Reviewer #1:

Remarks to the Author:

The manuscript of Shi and co-workers describes the palladium-catalyzed synthesis of di- and triaryl amines from aryl bromides and nitrogen. While such reactions are well preceded using ammonia or arylamines themselves, the reaction with dinitrogen is scarcely known. For example, non-substituted anilines have been prepared by titanium-nitrogen fixation complexes (reference 36 in the manuscript). Clearly, compared to this and other related work the present procedure is scientifically novel and justifies publication. Moreover, the work is clearly reported, and the experimental part is sound.

However, there are several problems associated with this work:

1. In the introduction (lines 36-50) the authors make the impression (intentionally?) that homogeneous catalysis offers potential to replace the current Haber-Bosch based chemistry. This is not the case!
2. The presented procedure does neither offer an economic nor ecologic advantage at present. This should be clearly stated. The authors use 4 equiv of lithium, 5 mol% Pd and costly phosphine ligands to make relatively simple aryl amines. All the products can be made more efficiently at present via Haber-Bosch based chemistry. In that respect, the presented procedure is not potentially applicable as stated in the conclusion (line 205).
3. Hence, in its present form I believe a publication in Nature Communication is too ambitious.
4. If the authors could demonstrate more convincingly the advantages of their procedure, publication might be considered again (e.g. preparation of functionalized and diverse N-labelled compounds).

A minor thing: arylbromides are usually written as aryl bromides

Reviewer #2:

Remarks to the Author:

The manuscript of Shi and coworkers presents the synthesis of a range of amines, N-containing heterocycles and polymers from N₂ in a two-step process involving initial reduction by lithium to form lithium nitride, then addition of aryl bromides and catalytic amounts of a Pd(0) source, a ligand, and other additives. Both steps occur at the relatively high temperature of 150 °C, so it is difficult to say exactly how the reaction is working. However, it seems clear that the mixture is effective. Their ¹⁵N labelling also suggests that the N comes from N₂ (although more evidence is needed, see below).

The reactivity of lithium nitride has only been explored to a limited extent, and no reliable, general and useful syntheses have been put forward. The reactions in this manuscript are a step in this direction. Whether or not this will lead to truly practical syntheses of amines is yet to be seen, but it is a definitely a very interesting possibility. This work presents a reaction that looks quite general and versatile and has been shown to make a wide variety of amines, thus I am quite positive about the science. However, the manuscript is somewhat messy in a number of places and full of errors, and there are a number of points that the authors should improve before publication. See below:

- The harsh conditions suggest the possibility that this process proceeds not via homogeneous catalysis but by other means, e.g. Pd nanoparticles. The authors should address this somewhere.
- The proton appearing in the secondary amine products is never explained. Do the authors have an idea of where it comes from?

- p2 "Current methods are also limited to active organoelectrophiles or valuable and strong reductants, which greatly limits their applications" - This seems to be in conflict with the current work, which also uses a strong reductant (lithium).

- p2: "druggable" - is this a word?

- Fig 1: Haber is spelled wrong. N-contained should be "N-containing". "catytic" - typo

- Fig 2: Equivalent amounts are given here of the reagents, however, what are they relative to? Are they relative to nitrogen? If so, how was this done practically, as it would involve exact amounts of N₂?

- Fig 2: It is not explained in B (or the caption) under what conditions this reaction takes place. Somewhere the conditions should be added (or the letter B. should be moved up so it encompasses the second reaction, then it would be clear).

- p7: I am not sure what is meant by "semi-one-pot" - to me it seems just like a one-pot procedure as there is no purification process between steps (as far as I can tell). The authors might mean something like "one-pot, two-step" instead.

- p10: "nitride had never been applied to C-N bond formation in organic synthesis" - this is not true, there are a number of literature reports on reactions of this type, although they are mostly from a long time ago (e.g. <https://doi.org/10.1021/jo01014a001>).

- SI: The spectra are not labelled so we don't know what kind of spectra they are.

- SI: No ¹⁵N NMR spectra of the labelled compounds are shown. This is very important as there are other potential sources of nitrogen in the reaction mixture, so we need to be absolutely sure that the ¹⁵N is present in the products.

- SI: All of the HR mass spectra need to be added to the SI for final proof of formula.

- SI: The references don't seem to be in Nature Comm. format.

Reviewer #3:

Remarks to the Author:

This paper reports a direct catalytic nitrogenation reaction to produce a variety of amine-containing compounds, diarylamines, triarylaminines, and heterocycles from organohalides using dinitrogen as the nitrogen source. This is highly valuable chemistry, and will therefore be of significant interest to the community, and, given the interest in nitrogenation reactions in the synthesis of new compounds, will also be of interest to the wider field.

Thus, it is the opinion of this referee this is a report that could belong in Nature Communications. That said, there are some significant concerns about the analysis and presentation of results, and therefore changes/further evidence is required before this referee can be convinced that this paper is suitable:

Figure 1A – the importance of nitrogen-containing molecules does not belong here. The authors do not even report the syntheses of these exemplar molecules in this paper; if that were the case, then maybe it should be included, but they do not. It is unnecessary and strange, and should be removed. Figure 1A is very busy, anyway, and this may also make the rest of the figure clearer.

The phrase "This represents the first successful and valuable catalytic transformation for preparing

N-containing organic compounds directly from dinitrogen, which has great potential for the synthesis of useful diarylamines" (line 107-109) is troublesome. Whilst this reaction appears to be superior to the precedent (even though there are several stoichiometric reagents in the reaction mixture on top of Pd salts etc), there have been catalytic syntheses from dinitrogen (see Figure 1). The authors should therefore not make such grandiose claims.

The reaction is performed under a nitrogen atmosphere. How do the authors know that the nitrogenation is coming from the Li_3N and not from the dinitrogen atmosphere through some other means? It should be simple to do some competitive labelling experiments between the Li salt and the atmosphere to confirm this. Furthermore, a KIE (although not as large for atoms such as N) for this reaction should also be examined. Kinetic data is needed for this reaction. Also, stoichiometric reactions may be useful here.

The general scheme in the ESI claims that 6 equiv of Li and $\text{Pd}(\text{dba})_2$ 0.025 mol% were used, but in the manuscript, the reaction Schemes say that 4 equiv of Li and 2.5 mol% $\text{Pd}(\text{dba})_2$ were used, with the later Scheme (Figure 3) saying that 0.025 equiv of $\text{Pd}(\text{dba})_2$ were used. The authors should check these discrepancies in the numbers and the units used and ensure their consistency, or split the general reaction method into several examples.

Put a figure caption by each NMR spectrum in the ESI. Also, some of the spectra presented appear to be contaminated. Label the contaminants.

A significant concern is with the mechanism presented in the ESI. Why is it buried in the ESI, with no reference to it in the main text? This is not the function of the ESI. Moreover, it is unclear how this mechanism was arrived at (with no experimental data or references provided in the text or ESI). Therefore, the authors must investigate the reaction fully by performing kinetics experiments and to probe salient reaction steps through theoretical calculations to determine the order of the reaction and the RDS. Stoichiometric reactions to attempt to isolate or identify reaction intermediates is relatively simple and should be attempted. Furthermore, the use of labelling (^{15}N to show the origin of the N atoms) and D labelling (e.g. the H_2O) to determine if these steps are involved in bond breaking/forming steps and where the labels are incorporated should also be undertaken. When this is accomplished, then the full description of the mechanism of the reaction, and how it was determined should be included in the main body of the text.

Reviewer: 1

1) In the introduction (lines 36-50) the authors make the impression (intentionally?) that homogeneous catalysis offers potential to replace the current Haber-Bosch based chemistry. This is not the case!

Answer: Thanks for the reviewer's comment. The corresponding modifications were added in the revised manuscript and highlight by yellow. Please refer to the marked version to check the revision as "As a long-term fundamental challenge, the activation and direct transformation of dinitrogen (N_2) has received much attention since the early twentieth century.⁷ The transformation of dinitrogen into ammonia through heterogeneous catalysis has a long history of application, and has been recognised as the most important reaction for promoting human's life.⁸ Recent advances have demonstrated the potential of homogeneous catalysis to produce ammonia directly from dinitrogen using complex catalysts in the presence of highly valuable reductants and proton sources.⁹ With great efforts by coordination chemists, various complexes with dinitrogen ligands have been prepared using almost all transition metals in different coordination modes in the last half century.^{10,11} By Coordination, dinitrogen in complexes was activated and has been used to prepare different N-containing organic compounds in a stoichiometric manner (Figure 1A).¹²⁻²² Despite some synthetic

cycles having been reported for the synthesis of N-containing molecules,²³⁻²⁶ to the best of our knowledge, catalytic transformation to carry out direct nitrogenation of organic molecules from dinitrogen (N₂) has not been approached up to date. Moreover, the use of active organoelectrophiles and strong reductants in synthetic cycles greatly limits their applications.”.

2) The presented procedure does neither offer an economic nor ecologic advantage at present. This should be clearly stated. The authors use 4 equiv of lithium, 5 mol% Pd and costly phosphine ligands to make relatively simple aryl amines. All the products can be made more efficiently at present via Haber-Bosch based chemistry. In that respect, the presented procedure is not potentially applicable as stated in the conclusion (line 205).

Answer: Thanks for the reviewer’s comment. The corresponding modifications were added in the revised manuscript in yellow color. Please refer to the marked version to check the revision as “ In summary, the potentially applicable one-pot/two-step protocol for direct catalytic nitrogenation by using dinitrogen as nitrogen source was first developed through Pd catalysis in the presence of Li as reductant. This chemistry was proved to occur through lithium nitride (Li₃N) as the key platform chemical. This is also the first example of Li₃N being used to construct new C–N bonds in organic synthesis. Not only does the method show that Li₃N is a potential platform chemical for the synthesis of N-containing molecules, but also allows direct nitrogenation to obtain N-containing molecules from dinitrogen while avoiding the traditional N₂-NH₃-HNO₃ procedure. However, the chemistry is still suffering from the relative harsh condition and the use of Li as reductants. Further efforts to explore other cheaper and safer reductants and expand the synthetic applications of useful chemicals are underway in our laboratory.”.

3) Hence, in its present form I believe a publication in Nature Communication is too ambitious. 4) If the authors could demonstrate more convincingly the advantages of their procedure, publication might be considered again (e.g. preparation of functionalized and diverse N-labelled compounds).

Answer: Thanks for the reviewer’s comment. Our procedure provides a successful pathway for catalytic nitrogenation to synthesize diverse and highly valuable N-

containing chemicals from dinitrogen, such as bi/triarylamines, carbazoles, tribenzylamine and pyrrole. Moreover, with this method, ^{15}N atoms were easily incorporated into organic molecules and structurally diversified polyanilines were also generated in one pot, showing great potential for materials chemistry.

A minor thing: arylbromides are usually written as aryl bromides.

Answer: Thanks for the reviewer's comment. According to the reviewer's suggestion, we have used "aryl bromides" instead of "arylbromides" in the article.

Reviewer: 2

The harsh conditions suggest the possibility that this process proceeds not via homogeneous catalysis but by other means, e.g. Pd nanoparticles. The authors should address this somewhere.

Answer: Thanks for the reviewer's constructive comments. According to the reviewer's suggestion, we have performed Hg (0) poisoning test and the yield showed that the reaction was inhibited in the presence of Hg (0) which was the typically catalytic characters of palladium nanoparticles. And this result has been added into the revised manuscript and SI.

The proton appearing in the secondary amine products is never explained. Do the authors have an idea of where it comes from?

Answer: Thanks for the reviewer's comment. In the General Procedure of SI, we have explained "The reaction mixture was allowed to cool to room temperature and quenched by water", The proton appearing in the secondary amine products comes from water when the reaction mixture was quenched.

p2 "Current methods are also limited to active organoelectrophiles or valuable and strong reductants, which greatly limits their applications" - This seems to be in conflict with the current work, which also uses a strong reductant (lithium).

Answer: Thanks for the reviewer's constructive comment. Compared to the common

strong and expensive reductants used in produce N-containing organic molecules directly from dinitrogen, lithium has relatively weaker reducibility and much cheaper and has been less reported. Definitely we are looking for better reductants in our system.

p2: "druggable" - is this a word?

Answer: Thanks for the reviewer's comment. We have deleted the "druggable" in revised manuscript. The corresponding modifications were added in the revised manuscript in yellow colour. Please refer to the marked version to check the revision as "N-containing molecules in drug design and polymeric materials greatly promote the quality of human beings' life.^{2,3}" and "we further expanded our chemistry to incorporate ¹⁵N atoms into privileged scaffolds in drug discovery (Figure 3D)".

Fig 1: Haber is spelled wrong. N-contained should be "N-containing". "cattytic" - typo

Answer: Thanks for the reviewer's comment. We have checked the Fig 1 carefully and corrected all the mistakes. We highlight all changes in blue.

Fig 2: Equivalents are given here of the reagents, however, what are they relative to? Are they relative to nitrogen? If so, how was this done practically, as it would involve exact amounts of N₂?

Answer: Thanks for the reviewer's comment. Equivalents given in Fig 2 of the reagent are relative to aryl bromide substrates, not relative to nitrogen. We have not involved exact amounts of N₂, which was only used in 1 atmosphere.

Fig 2: It is not explained in B (or the caption) under what conditions this reaction takes place. Somewhere the conditions should be added (or the letter B. should be moved up so it encompasses the second reaction, then it would be clear).

Answer: Thanks for the reviewer's comment. According to the reviewer's suggestion, we have moved the letter B up as suggested.

p7: I am not sure what is meant by "semi-one-pot" - to me it seems just like a one-pot

procedure as there is no purification process between steps (as far as I can tell). The authors might mean something like "one-pot, two-step" instead.

Answer: Thanks for the reviewer's comment. According to the reviewer's suggestion, we have checked the article and have used "one-pot/two-step" instead of "semi-one-pot".

p10: "nitride had never been applied to C - N bond formation in organic synthesis" - this is not true, there are a number of literature reports on reactions of this type, although they are mostly from a long time ago (e.g. <https://doi.org/10.1021/jo01014a001>).

Answer: Thanks for the reviewer's comment. According to the reviewer's suggestion, the corresponding reference was added as ref. 51. We used "nitrides have rarely been applied to C-N bond formation" instead of "nitride had never been applied to C-N bond formation in organic synthesis".

SI: The spectra are not labelled so we don't know what kind of spectra they are.

Answer: Thanks for the reviewer's comment. According to the reviewer's suggestion, all the spectra have been labelled in SI.

SI: No ¹⁵N NMR spectra of the labelled compounds are shown. This is very important as there are other potential sources of nitrogen in the reaction mixture, so we need to be absolutely sure that the ¹⁵N is present in the products.

Answer: Thanks for the reviewer's comment. According to the reviewer's suggestion, we have performed ¹⁵N NMR spectra of the labelled compounds (¹⁵N-5a and ¹⁵N-17) and put the spectra in the SI (please see Supplementary Figure 94 and Supplementary Figure 97).

SI: All of the HR mass spectra need to be added to the SI for final proof of formula.

Answer: Thanks for the reviewer's comment. According to the reviewer's suggestion, all of the HR mass spectra have been added to the SI for final proof of formula.

SI: The references don't seem to be in Nature Comm. format.

Answer: Thanks for the reviewer's comment. According to the reviewer's suggestion, all the references in SI have been changed to the format for Nat. Comm..

Reviewer: 3

Figure 1A - the importance of nitrogen-containing molecules does not belong here. The authors do not even report the syntheses of these exemplar molecules in this paper; if that were the case, then maybe it should be included, but they do not. It is unnecessary and strange, and should be removed. Figure 1A is very busy, anyway, and this may also make the rest of the figure clearer.

Answer: Thanks for the reviewer's comment. As the reviewer stated above, we have removed the primary Figure 1A.

The phrase "This represents the first successful and valuable catalytic transformation for preparing N-containing organic compounds directly from dinitrogen, which has great potential for the synthesis of useful diarylamines" (line 107-109) is troublesome. Whilst this reaction appears to be superior to the precedent (even though there are several stoichiometric reagents in the reaction mixture on top of Pd salts etc), there have been catalytic syntheses from dinitrogen (see Figure 1). The authors should therefore not make such grandiose claims.

Answer: Thanks for the reviewer's comment. Please refer to the marked version to check the revision as "This is a successful and valuable catalytic transformation for preparing N-containing organic compounds directly from dinitrogen, which exhibited a great potential for the synthesis of useful diarylamines."

The reaction is performed under a nitrogen atmosphere. How do the authors know that the nitrogenation is coming from the Li_3N and not from the dinitrogen atmosphere through some other means? It should be simple to do some competitive labelling experiments between the Li salt and the atmosphere to confirm this. Furthermore, a KIE (although not as large for atoms such as N) for this reaction should also be examined.

Kinetic data is needed for this reaction. Also, stoichiometric reactions may be useful here.

Answer: Thanks for the reviewer's comment. According to the reviewer's suggestion, we have performed ^{15}N -labelling experiments. In one-pot/two-step, we have prepared Li_3^{15}N in step I, and then in step II under argon atmosphere and dinitrogen atmosphere, respectively. Both got ^{15}N -labelling product and no ^{14}N -labelling product obtained, indicating that nitrogenation is coming from the Li_3N and not from the dinitrogen atmosphere through some other means. These results were added into the SI (please see Supplementary Table 3).

The general scheme in the ESI claims that 6 equiv of Li and $\text{Pd}(\text{dba})_2$ 0.025 mol% were used, but in the manuscript, the reaction Schemes say that 4 equiv of Li and 2.5 mol% $\text{Pd}(\text{dba})_2$ were used, with the later Scheme (Figure 3) saying that 0.025 equiv of $\text{Pd}(\text{dba})_2$ were used. The authors should check these discrepancies in the numbers and the units used and ensure their consistency, or split the general reaction method into several examples.

Answer: Thanks for the reviewer's comment. According to the reviewer's comment, we have checked the manuscript and ESI carefully. We have unified content and highlight in blue in revised manuscript.

Put a figure caption by each NMR spectrum in the ESI. Also, some of the spectra presented appear to be contaminated. Label the contaminants.

Answer: Thanks for the reviewer's comment. According to the reviewer's comment, we have put a figure caption highlight in yellow by each NMR spectrum in the SI and replaced the contaminated spectrum with purified spectrum.

A significant concern is with the mechanism presented in the ESI. Why is it buried in the ESI, with no reference to it in the main text? This is not the function of the ESI. Moreover, it is unclear how this mechanism was arrived at (with no experimental data or references provided in the text or ESI). Therefore, the authors must investigate the reaction fully by performing kinetics experiments and to probe salient reaction steps through theoretical

calculations to determine the order of the reaction and the RDS. Stoichiometric reactions to attempt to isolate or identify reaction intermediates is relatively simple and should be attempted. Furthermore, the use of labelling (^{15}N to show the origin of the N atoms) and D labelling (e.g. the H_2O) to determine if these steps are involved in bond breaking/forming steps and where the labels are incorporated should also be undertaken. When this is accomplished, then the full description of the mechanism of the reaction, and how it was determined should be included in the main body of the text.

Answer: Thanks for the reviewer's comment. To response the reviewer's concern, we have moved the full description of the mechanism of the reaction in the main body of the text. We made our full efforts by experiments, but we could not achieve any intermediates at this stage. We have used labelling ^{15}N to show the origin of the N atoms is from the Li_3N (please see Supplementary Table 3) and D labelling to determine the desired product (please see HRMS Spectrum of Supplementary Figure 192) was produced by hydrolysis. These results and related conclusions were added into the revised manuscript and SI.

Reviewers' Comments:

Reviewer #2:

Remarks to the Author:

The authors have addressed my comments very well, and appear also to have addressed those of the other reviewers. I find the manuscript to be very nice now (apart from some issues below), and on the level expected for a journal such as this. Overall, I do not agree with the first reviewer who does not find the results suitable for Nature Communications - I find the results to be very interesting and indeed exciting as they are very different from the common approach to N₂ functionalization (i.e. using early TMs). However, some issues remain that need to be cleared up.

The only major problem is in regard to the mechanism. In response to my comment about nanoparticles, the authors performed reactions with mercury and determined that nanoparticles are likely present and at least some of the reactivity is due to these particles. Thus, I do not understand why a complicated mechanism for catalysis on a (homogeneous) Pd species is now added to the main text, when the mercury experiment suggests that this is unlikely. I don't think it is necessary for the authors to speculate on the mechanism, especially considering the uncertainties about nanoparticles. In any case, I don't think the manuscript needs mechanistic discussion, as the synthetic results speak for themselves. I recommend removing all of the discussion of the mechanism, and perhaps expansion of the discussion of the Hg reaction and its consequences.

Minor points:

- Abstract: "Herein, we reported an elegant example of..." - referring to your own work as "elegant" should be avoided.
- The English in the manuscript is quite bad still - it should be carefully edited by someone.
- Some of the schemes in the SI seem to have been messed up somehow, the words are jumbled up.

Reviewer #3:

Remarks to the Author:

The authors have made a number of changes required by the reviewers, and have performed additional experiments. I have concerns regarding the mechanism in the manuscript:

Kinetics have not been undertaken by the authors, as suggested, which may help with the mechanism.

The Hg(0) poisoning experiments seem to imply that Pd nanoparticles are an active catalytic species, but then are only provided as an off-cycle process. This is problematic for two reasons:

1. Hg(0) poisoning experiments have been demonstrated to be inadequate in the study of reaction mechanism (e.g. ACS Catal. 2019, 9, 2984–2995) and therefore other poisoning experiments need to be undertaken to verify this.
2. Is there any evidence for the Pd(0) nanoparticles being an off-cycle species? The role of the nanoparticles and indeed the PdLn species in the reaction needs to be verified (kinetics may help)

Reviewer: 2

1) The only major problem is in regard to the mechanism. In response to my comment about nanoparticles, the authors performed reactions with mercury and determined that nanoparticles are likely present and at least some of the reactivity is due to these particles. Thus, I do not understand why a complicated mechanism for catalysis on a (homogeneous) Pd species is now added to the main text, when the mercury experiment suggests that this is unlikely. I don't think it is necessary for the authors to speculate on the mechanism, especially considering the uncertainties about nanoparticles. In any case, I don't think the manuscript needs mechanistic discussion, as the synthetic results speak for themselves. I recommend removing all of the discussion of the mechanism, and perhaps expansion of the discussion of the Hg reaction and its consequences.

Answer: Thanks for the reviewer's comment. According to the reviewer's suggestion, we have removed all of the discussion of the mechanism of the manuscript and expanded the discussion of the Hg reaction and its consequences in the manuscript. We have highlight in blue in the revised manuscript.

2) Abstract: "Herein, we reported an elegant example of..." - referring to your own work as "elegant" should be avoided.

Answer: Thanks for the reviewer's comment. According to the reviewer's suggestion, we have used "novel" instead of "elegant" in the revised manuscript.

3) . The English in the manuscript is quite bad still - it should be carefully edited by someone.

Answer: Thanks for the reviewer's comment. According to the reviewer's suggestion, we have got Simon Partridge, PhD, from Liwen Bianji, Edanz

Editing China (www.liwenbianji.cn/ac) to edit this manuscript carefully.

4) Some of the schemes in the SI seem to have been messed up somehow, the words are jumbled up.

Answer: Thanks for the reviewer's comment. We have successfully converted the SI document into PDF format again, and solved all format problems.

Reviewer: 3

Kinetics have not been undertaken by the authors, as suggested, which may help with the mechanism. The Hg(0) poisoning experiments seem to imply that Pd nanoparticles are an active catalytic species, but then are only provided as an off-cycle process. This is problematic for two reasons:

1. Hg(0) poisoning experiments have been demonstrated to be inadequate in the study of reaction mechanism (e.g. ACS Catal. 2019, 9, 2984-2995) and therefore other poisoning experiments need to be undertaken to verify this.
2. Is there any evidence for the Pd(0) nanoparticles being an off-cycle species? The role of the nanoparticles and indeed the PdLn species in the reaction needs to be verified (kinetics may help).

Answer: Thanks for the reviewer's constructive comments. According to the literature (J. Am. Chem. Soc. **2003**, 125, 10301), "Hg(0) is probably most effective in poisoning metals that form an amalgam, such as Pt, Pd, and Ni". According to the other literature (Adv. Synth. Catal. **2010**, 352, 33), "The usefulness of Mercury Test is still rather controversial concerning metals that do not form an amalgam with mercury (Ir, Rh, Ru), and also in the reactions using low-oxidation state naked ligand-free Pd(0)." In our work, we used Pd₂(dba)₃ as catalyst and RuPhos/Ad₂PⁿBu as ligand. The catalyst Pd is not naked ligand-free, so we think Mercury Test is probably most effective in our work (Chem. Eur. J. **2015**, 21, 1578; J. Am. Chem. Soc. **1998**, 120, 5653). Also, to following the reviewer 2 and the editor's comments, we removed the

mechanistic studies in this final version.